# NEURAL PROBABILISTIC MOTOR PRIMITIVES FOR HUMANOID CONTROL

**Josh Merel,** * **Leonard Hasenclever,** * **Alexandre Galashov,**
**Arun Ahuja, Vu Pham, Greg Wayne, Yee Whye Teh, & Nicolas Heess**
DeepMind
London, UK
`{jsmerel,leonardh,agalashov,arahuja,vuph,`
`gregwayne,ywteh,heess}@google.com`

## ABSTRACT

We focus on the problem of learning a single motor module that can flexibly express a range of behaviors for the control of high-dimensional physically simulated humanoids. To do this, we propose a motor architecture that has the general structure of an inverse model with a latent-variable bottleneck. We show that it is possible to train this model entirely offline to compress thousands of expert policies and learn a motor primitive embedding space. The trained *neural probabilistic motor primitive* system can perform one-shot imitation of whole-body humanoid behaviors, robustly mimicking unseen trajectories. Additionally, we demonstrate that it is also straightforward to train controllers to reuse the learned motor primitive space to solve tasks, and the resulting movements are relatively naturalistic. To support the training of our model, we compare two approaches for offline *policy cloning*, including an experience efficient method which we call *linear feedback policy cloning*. We encourage readers to view a supplementary video summarizing our results.

## 1 INTRODUCTION

A broad challenge in machine learning for control and robotics is to produce policies capable of general, flexible, and adaptive behavior of complex, physical bodies. To build policies that can effectively control simulated humanoid bodies, researchers must simultaneously overcome foundational challenges related to high-dimensional control, body balance, and locomotion. Recent progress in deep reinforcement learning has raised hopes that such behaviors can be learned end-to-end with minimal manual intervention. Yet, even though significant progress has been made thanks to better algorithms, training regimes, and computational infrastructure, the resulting behaviors still tend to exhibit significant idiosyncrasies (e.g. Heess et al., 2017; Bansal et al., 2018).

One advantage of working with humanoids in this context is that motion capture data is widely available and can serve to help design controllers that produce apparently humanlike movement. Indeed, recent developments are now allowing for the production of highly specialized expert policies which robustly, albeit narrowly, reproduce single motion capture clips (e.g. Liu et al. (2010); Peng et al. (2018)).

A remaining challenge on the way to truly flexible and general purpose control is to be able to sequence and generalize individual movements or "skills" in a task-directed manner. Achieving this goal requires not just the ability to acquire individual skills in the first place, but also an architecture and associated training procedure that supports representation, recruitment, and composition of a large number of skills.

This paper presents a step in this direction. Specifically, the setting we focus on will be one in which we have a large number of robust experts that perform single skills well and we wish to transfer these skills into a shared policy that can do what each expert does as well as the expert, while also generalizing to unseen behaviors within the distribution of skills. To this end we design a system

---

*Equal contribution.

that performs one-shot imitation as well as permits straightforward reuse (or transfer) of skills. We require our approach to scale to a very large number of individual skills while also keeping manual intervention and oversight to a minimum.

Our primary contribution is the development of a neural network architecture that can represent and generate many motor behaviors, which we refer to as *neural probabilistic motor primitives*. This architecture is designed to perform one-shot imitation, while learning a dense embedding space of a large number of individual motor skills. Once trained, this module does not just reproduce individual behaviors in the training data, but can sequence and compose these behaviors in a controlled fashion as well as synthesize novel movements consistent with the training data distribution. Empirically, we also find that training controllers to reuse this learned motor primitive module for new tasks generates surprisingly human-like movement and the behavior generated seems to interpolate the space of behaviors well.

In order to facilitate transfer and compression of expert skills at the scale of thousands of behaviors, we wish to avoid closed-loop RL training. We call the general, offline, functional transfer of policy content *policy transfer* or *policy cloning* and consider two approaches. The natural baseline approach involves the application of behavioral cloning to data gathered by executing experts many times, with noise, and logging intended expert actions, resembling the approach of Laskey et al. (2017). This works well, as it ensures the student behaves like the expert not only along nominal expert rollouts but also at points arrived at by perturbing the expert. However, this approach may require many rollouts, which can be costly to obtain in many settings. As a more efficient alternative we therefore consider a second solution that operates by comprehensively transferring the functional properties of an expert to a student policy by matching the local noise-feedback properties along one or a small number of representative expert reference trajectories. We call this specific proposal *linear feedback policy cloning (LFPC)*, and we demonstrate that it is competitive with behavioral cloning from many more rollouts in our setting.

## 1.1 BACKGROUND & RELATED WORK

Recent efforts in RL for *humanoid control* build on a large body of research in robotics and animation. While contemporary results for learning from scratch (Schulman et al., 2015; Heess et al., 2017) can be impressive the behaviors are not consistently human-like. Learning from motion capture (mocap) can provide strong constraints, especially for running (Peng et al., 2017; Merel et al., 2017). Several recent approaches have demonstrated that it is possible to acquire specific behavioral skills, possibly jointly with external RL objectives (Merel et al., 2017; Peng et al., 2018; Liu & Hodgins, 2018). At present, the policies produced tend to be restricted to single skills/behaviors and can require very large quantities of environment interactions, motivating us to seek methods which reuse existing single-skill expert policies.

*Knowledge transfer* refers to the broad class of approaches which transfer the input-output functional mapping, to some extent or another, from a *teacher* (or *expert*) to a *student* (Hinton et al., 2015; Srinivas & Fleuret, 2018; Furlanello et al., 2018). *Distillation* connotes the transfer of function from one or more expert systems into a single student system often with the goal of compression or of combining multiple experts qualities (Hinton et al., 2015; Parisotto et al., 2015; Rusu et al., 2015; Teh et al., 2017). *Imitation learning* is the control-specific term for the production of a student policy from either an expert policy or the behavioral demonstrations of an expert. One basic algorithm is *behavioral cloning*, which refers to supervised training of the policy from state-action pairs. In the most simple case it only requires examples from the expert. A broader setting is that in which more liberal queries to the expert are permitted; e.g. for the online-imitation setting as in DAGGER (Ross et al., 2011). This setting is often satisfied e.g. if we wish to combine behavior from multiple experts.

*One-shot imitation* is a concept which means that a trained system, at test time, can watch an example behavior and imitate it, as, for instance, in Duan et al. (2017). More similar to our work is the setting examined by Wang et al. (2017), in which full-body humanoid movements were studied. Compared with this latter work, we will employ an architecture here that encourages imitation of motor details, rather than overall movement type, and we scale our approach to more expert demonstrations. The most similar work also demonstrates large-scale one-shot humanoid tracking and was contemporaneously published (Chentanez et al., 2018); the approach they described involves direct tracking as well as failure recovery, but relative to our work the authors do not consider skill reuse.

The notion of *motor primitives* is widespread in neuroscience, where there is evidence that lower dimensional control signals can selectively coordinate and blend behaviors produced by spinal circuits (Bizzi et al., 2008), and that the cortex organizes the space of primitive motor behaviors (Graziano, 2006). In our setting, motor primitives refer to the reusable embedding space learned from many related behaviors and the associated context-modulable policy capable of generating sensory-feedback-stabilized motor behavior when executed in an environment. The particular architecture we consider is inspired by the formalization presented in Todorov & Ghahramani (2003), which places a probabilistic latent bottleneck on the sensory-motor mapping.

In the robotics literature, there is a rich line of research into various parameterizations of motion trajectories used for robot control. A class of these are referred to as "movement primitives" (e.g. Schaal et al., 2003), including the "probabilistic movement primitives" of Paraschos et al. (2013) (see also e.g. Neumann et al., 2014). These approaches can be seen as specific implementation choices for a certain notion of motor primitive, which emphasize the parameterization and learning of movement trajectories from repeated demonstrations (Paraschos et al., 2013; Meier & Schaal, 2016), rather than learning the actuation/stabilization element, which is often handled by a pre-specified PID controller.

It has previously been recognized that linear-feedback policies can work well around optimal trajectories or limit cycles even for high DoF bodies. These can be obtained by sample-based optimization (e.g. Ding et al. (2015)) or by differential dynamic programming (Morimoto & Atkeson, 2003; Tassa et al., 2012; 2014). For linear-quadratic-Gaussian control (Athans, 1971) or differential dynamic programming (Mayne, 1966; Jacobson & Mayne, 1970), we obtain feedback policies where the feedback terms are computed from the value function, amounting effectively to feedback-stabilized plans. Work by Mordatch et al. (2015) has shown that linear-feedback policies resulting from trajectory optimization can be used to train neural networks. We employ a similar idea to transfer optimal behavior from an existing policy, observing that an optimal policy implicitly reflects the structure of the (local) value landscape and appropriately functions as a feedback controller.

## 2 TRANSFER AND COMPRESSION OF EXPERT BEHAVIORS

In this section, we will first briefly describe the expert policies used in this work (Sec. 2.1). We then describe the Neural Probabilistic Motor Primitive architecture and objective (Sec. 2.2). We then describe two approaches for training the module offline (Sec. 2.3).

### 2.1 OBTAINING EXPERTS FROM MOTION CAPTURE DATA

In order to study how to transfer and consolidate experts, we must be able to generate adequate quantities of expert data. For this work, we use expert policies trained to reproduce motion capture clips. The approach we use for producing experts is detailed more fully in Merel et al. (2018) and largely follows Peng et al. (2018). It yields time-indexed neural network policies that are robust to moderate amounts of action noise (see appendix A for additional details on the training procedure). Some examples of the resulting single-skill time-indexed policies that are obtained from this procedure are

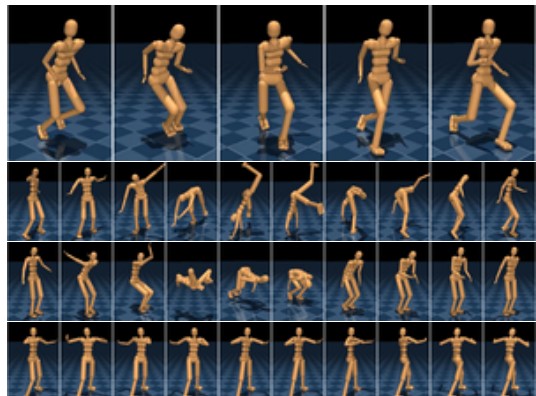

Figure 1: Examples of representative experts learned from motion capture. From top to bottom, these are "run and dodge", "cartwheel", "backflip", and "twist". See accompanying video. Note that these four behaviors will be used as representative examples for validation in single-skill transfer experiments.

depicted in Fig. 1. All our experts were trained in MuJoCo environments (Todorov et al., 2012).

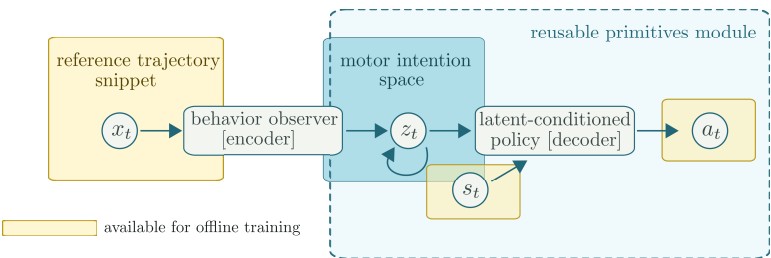

Figure 2: Neural probabilistic motor primitive architecture for one-shot skill deployment. The yellow-highlighted information are available for offline, supervised training. Once the full model has been learned, the decoder can be reused as a policy in other settings.

**Data** We use the CMU Mocap database[1], which contains more than 2000 clips of varying lengths from more than 100 subjects. The motions in this dataset are quite varied, including many clips of walking, turning, running, jumping, dancing, various hand movements, and many more idiosyncratic behaviors. From this, we selected various clips of generic whole-body movements – any clips longer than 6 seconds were cut into smaller pieces yielding approximately 3000, roughly 2-6 second snippets. Just over half of these are generic locomotion such as walking, running, jumping and turning. The rest of the clips mostly contained diverse hand movements while standing. We trained one expert policy per selected snippet, yielding 2707 expert policies in our training set.

## 2.2 Neural Probabilistic Motor Primitives

Our goal is to obtain a motor primitive module that can flexibly and robustly deploy, sequence, and interpolate a diverse set of skills from a large database of reference trajectories without any manual alignment or other processing of the raw experts. This requires a representation that does not just reliably encode all behavioral modes but also allows effective indexing of behaviors for recall. To ensure plausible and reliable transitions it is further desirable that the encoding of similar behaviors should be close in some sense in the representation space.

**Compression of many expert skills via a latent variable inverse model** We achieve this goal by training an autoregressive latent variable model of the state-conditional action sequence which, at training time, is conditioned on short look-ahead snippets of the nominal/reference trajectory (see Fig. 2). This architecture has the general structure of an inverse model, which produces actions based on the current state and a target. The architecture and training scheme are designed for the embedding space to reflect short-term motor behavior. As we demonstrate below, this allows for the selective execution of particular behavioral modes and also admits one-shot imitation via the trajectory encoder.

We use a model with a latent variable $z_t$ at each time step, modelling the state conditional action distribution. The encoder and decoder are distributions $q(z_t|z_{t-1}, x_t)$ and $\pi(a_t|z_t, s_t)$ where $s_t$ is the state as in preceding sections and $x_t$ is concatenation of a small number of future states $x_t = [s_t, ..., s_{t+K}]$. The encoder and decoder are MLPs with two and three layers, respectively. For architecture and experimental details see appendix B. The generative part of the model is given by:

$$p(a_{1:T}, z_{1:T}|s_{1:T}) = \prod_{t=1}^{T} p_z(z_t|z_{t-1})\pi(a_t|z_t, s_t). \tag{1}$$

Temporally nearby trajectory snippets should have a similar representation in the latent space. To implement this intuition, we choose an AR(1) process as a weak prior:

$$z_t = \alpha z_{t-1} + \sigma\epsilon, \ \ \epsilon \sim \mathcal{N}(0, I), \tag{2}$$

where $\sigma = \sqrt{1 - \alpha^2}$, ensuring that marginally $z_t \sim \mathcal{N}(0, I)$, and set $\alpha = 0.95$ in experiments unless otherwise stated. In subsequent efforts, it may be interesting to investigate different values of $\alpha$ and learnable priors.

---

[1]The CMU motion capture database is available at mocap.cs.cmu.edu.

In order to train this model, we consider the evidence lower bound (ELBO):

$$\mathbb{E}_q \left[ \sum_{t=1}^{T} \log \pi(a_t | s_t, z_t) + \beta \big( \log p_z(z_t | z_{t-1}) - \log q(z_t | z_{t-1}, x_t)) \right], \tag{3}$$

with a $\beta$ parameter to tune the weight of the prior. For $\beta = 1$ this objective forms the well-known variational lower bound to $\log p(a_{1:T} | s_{1:T})$. This objective can be optimized using supervised learning (i.e. behavioral cloning from noisy rollouts) offline.

Note we chose not to condition the encoder on actions, since we are interested in one-shot imitation in settings where actions are unobserved. We experimented with different values of $K$ and obtained similar performance. All the results reported in this paper use $K = 5$.[2]

Our architecture effectively implements a conditional information bottleneck between the desired future trajectory $x_t$ and the action $a_t$ given the past latent state $z_{t-1}$ (similar to Alemi et al. (2017)). As discussed above the auto-correlated prior encourages an encoding in which temporally nearby latent states from the same trajectory tend to be close in the latent space, and the information bottleneck more generally encourages a limited dependence on $x_t$ with $z_t$ forming a compressed representation of the future trajectory as required for the action choice.

### 2.3 TRAINING A STUDENT POLICY FROM A SET OF EXAMPLES

When transferring knowledge from an expert policy to a student we would like the student to replicate the expert's behavior in the full set of states plausibly visited by the expert. In our case, experts trained to reproduce single clips can be conceptualized as nonlinear feedback controllers around a nominal trajectory, and the manifold of states visited by experts can be thought of as a tube around that reference. We require the student to be able to operate successfully in and remain close to this tube even in the face of small perturbations.

Formally, to ensure that the student retains expert robustness, we would like expert actions $\mu_E(s)$ and student actions $\mu_\theta(s)$ to be close under a plausible (noisy) expert state distribution $\rho_E$. A surrogate loss used in imitation learning as well as knowledge transfer is the quadratic loss between actions (Ross et al., 2011) (or activations Srinivas & Fleuret (2018)).

$$\min_\theta \mathbb{E}_{s \sim \rho_E}[(\mu_E(s) - \mu_\theta(s))^2] \tag{4}$$

Behavioral cloning can refer to optimization of this objective, where $\rho_E$ is replaced with an empirical distribution of a set of state-action pairs $\mathcal{S}$. This works well if $\mathcal{S}$ adequately covers the state distribution later experienced by the student. Anticipating and generating an appropriate set of states on which to train the student typically requires many rollouts and can thus be expensive.

Since we are aiming to compress the behavior of thousands of experts we desire a computationally efficient method. We investigate two schemes that allow us to record the experts' state-action mappings on a small-sample estimate of the experts' state distributions and to then train the student via supervised learning. Both schemes are convenient to implement in a regular supervised learning pipeline and require neither querying many experts simultaneously (which limits scalability when dealing with thousands of experts) nor execution of the student at training time.

**Behavioral cloning from noisy rollouts** The first approach amounts to simply gathering a number of noisy trajectories from the expert (either under a stochastic policy or with noise injection) while logging the optimal/mean action of the expert instead of the noisy action actually executed. A version of this is equivalent to the DART algorithm of Laskey et al. (2017). We then perform behavioral cloning from that data.

Specifically, given an expert policy $\pi_E$, let $\mu_E(s)$ be the mean action of the expert in state $s$. To obtain noisy rollouts, we run $\pi_E^\eta$, the expert with moderate action noise ($\eta$) to obtain a set of data $\{s_k^\eta, \mu_k\}_{1...K}$, where $\mu_k = \mu_E(s_k^\eta)$. And we optimize the policy according to Eqn. 4, with the expectation over $s \sim \rho_E$ being approximated by a sum over the set of state and expert-actions

---

[2]We also experimented with ways to look further into the future by conditioning on $x_t = [s_t, s_{t+1}, s_{t+3}, s_{t+6}, s_{t+10}, s_{t+15}]$. This gave broadly similar results.

collected. While we expect this approach can work well, we do not expect it to be particularly efficient insofar as the expert may need to be executed for many rollouts.

**Linear-feedback policy cloning (LFPC)** The second approach, which we refer to as linear-feedback policy cloning (LFPC), logs the action-state Jacobian as well as the expert action along a single nominal trajectory. The Jacobian can be used to construct a linear feedback controller which gives target actions in nearby perturbed states during training (described below). This approach is not intended to outperform behavioral cloning, as this should not be possible for arbitrary quantities of expert rollout data. Instead the motivation for LFPC is to do as well as behavioral cloning while using considerably fewer expert rollouts.

As pointed out above, experts trained to reproduce single clips robustly can be thought of as non-linear feedback controllers around this nominal trajectory. The nominal trajectory refers to the sequence of nominal state-action pairs $\{s_t^\star, a_t^\star\}_{1\ldots T}$ obtained by executing $\mu_E(s)$ recursively from an initial point $s_0^\star$. Since expert behavior in our setting is well characterized by single nominal trajectories, we expect we can capture the relevant behavior of the expert by a linearization around the nominal trajectory[3].

Let $\delta s$ be a small perturbation of the state and let $\boldsymbol{J} = \frac{d\mu_E(s)}{ds}|_{s=s}$ be the Jacobian. Then

$$\mu_E(s + \delta s) = \mu_E(s) + \boldsymbol{J}\delta s + O\left(\|\delta s\|^2\right). \tag{5}$$

This linearization induces a linear-feedback-stabilized policy that at each time-step has a nominal action $a_t^\star$, but also expects to be in state $s_t^\star$, and correspondingly adjusts the nominal action with a linear correction based on discrepancy between the nominal and actual state at time $t$:

$$\mu_{FB}(s_t) = a_t^\star + \boldsymbol{J}_t^\star(s_t - s_t^\star), \quad \text{where} \quad \boldsymbol{J}_t^\star = \frac{d\mu_E(s)}{ds}\bigg|_{s=s_t^\star}. \tag{6}$$

We empirically validated that a linear feedback policy about the nominal trajectory of the expert can approximate the expert behavior reasonably well for clips we examine (see results Fig. 3).

Above we presented the expert as a feedback controller operating in a tube around some nominal trajectory with states $s_1^\star, \ldots, s_T^\star$, actions $a_1^\star, \ldots, a_T^\star$, and Jacobians $\boldsymbol{J}_1^\star, \ldots, \boldsymbol{J}_T^\star$. We approximate $\rho_E$ with the distribution of states introduced by state perturbations around this nominal trajectory:

$$\min_\theta \frac{1}{T} \sum_i \mathbb{E}_{\delta s_i \sim \Delta(s)}[\|\mu_E(s_i + \delta s_i) - \mu_\theta(s_i + \delta s_i)\|^2]. \tag{7}$$

However, this objective still requires expert evaluations at the perturbed states. Using the linearization described above we can replace the expert action $\mu_E(s + \delta s)$ with the Jacobian-based linear-feedback policy $\mu_{FB}(s + \delta s)$, which is available offline. This yields the LFPC objective:

$$\min_\theta \frac{1}{T} \sum_i \mathbb{E}_{\delta s_i \sim \Delta(s)}[\|\mu_\theta(s_i^\star + \delta s_i) - a_i^\star - \boldsymbol{J}_i^\star \delta s_i\|_2^2]. \tag{8}$$

One potentially important choice is the perturbation distribution $\Delta(s)$. Ideally, we would like $\Delta(s)$ to be the state-dependent distribution induced by physically plausible transitions, but estimating this distribution may require potentially expensive rollouts which we are trying to avoid. A cheaper object to estimate is the *stationary* transition noise distribution induced by noisy actions, which can be efficiently approximated from a small number of trajectories. Empirically, we found the objective 8 to be relatively robust to some variations in $\Delta$, and we use a fixed marginal distribution for all clips.

Objective 8 bears interesting similarities to approaches such as denoising autoencoders (Vincent et al., 2008), where networks can learn to ignore local noise perturbations on inputs sampled from a high-dimensional noise distribution. Further, Mordatch et al. (2015) successfully distill feedback policies obtained from a planner. One question left open by this latter work is that of how much data might be required. Empirically we show in the experiments below that the augmented objective 8 can produce the desired robustness even from a very limited set of states.

---

[3]Note that in contemporary neural network languages, it is straightforward to automatically compute the Jacobian of the actions with respect to observation inputs.

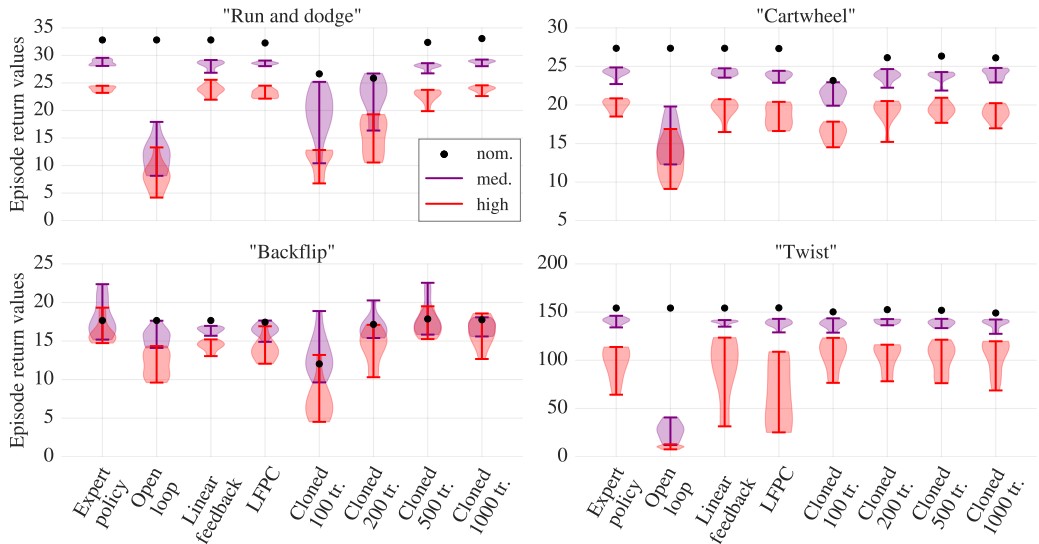

Figure 3: Comparisons of trajectory rollouts for 4 reference behaviors for the nominal trajectory and at varying noise levels. Note that the score is determined by similarity to motion-capture reference and the expert may be slightly suboptimal so slight improvements on the expert may arise by chance.

There are multiple, relevant perspectives on LFPC. From one perspective, LFPC amounts to a data augmentation method. From another vantage, the approach attempts to match the mean action as well as the Jacobian at the set of relevant behavioral states, here sampled along the nominal trajectory. In settings where expert behavior is more diverse or multimodal, LFPC should be applied to states which representatively cover relevant behavioral modes or perhaps are expanded backwards from goal states (roughly similar to the procedure used to expand LQR-trees by Tedrake 2009). Explicit Jacobian matching has been proposed elsewhere, for example in Czarnecki et al. (2017). See appendix C for further disambiguation relative to other approaches.

To train our Neural Probabilistic Motor Primitive architecture using LFPC we can adapt the objective in Eqn. 3 as follows:

$$\mathbb{E}_{\delta s, q}\left[\sum_{t=1}^{T} \log \pi(a_t + \boldsymbol{J}_t \delta s_t | s_t + \delta s_t, z_t) + \beta\big(\log p_z(z_t | z_{t-1}) - \log q(z_t | z_{t-1}, x_t + \delta x_t))\right], \quad (9)$$

where $\delta s_t$ are i.i.d. perturbations drawn from suitable perturbation distribution $\Delta$ and $\delta x_t$ is the concatenation of $[\delta s_t, \delta s_{t+1}, ..., \delta s_{t+K}]$.

## 3 EXPERIMENTS

### 3.1 VALIDATION: TRANSFER OF SINGLE-BEHAVIOR POLICIES

To ground our results in a simple setting, we begin with transfer of a single-skill, time-indexed policy from one network to another. We compare the performance of various time-indexed policies for each of the experts depicted in Fig. 1. We compare the original expert policy, an open-loop action sequence along the experts nominal (i.e. mean) trajectory, a linear feedback policy along the expert nominal trajectory, as well as the network trained to match the linear-feedback behavior (LFPC). In addition we compare to policies trained from 100, 200, 500 or 1000 trajectories with behavioral cloning. We compare each approach with no action noise, small action noise, and moderate action noise (noise is i.i.d. normal per actuator with standard deviation magnitude .05 and .1 respectively, for action ranges normalized to $[-1, 1]$). Note that, open loop control almost always fails if the state is perturbed by even a small $\epsilon$ (though perhaps surprisingly, the backflip can almost be executed open loop due to limited ground contact). Remarkably, LFPC with a single trajectory performs on par with behavioral cloning based on hundreds of trajectories (see Fig. 3). For additional validation, see appendix D.

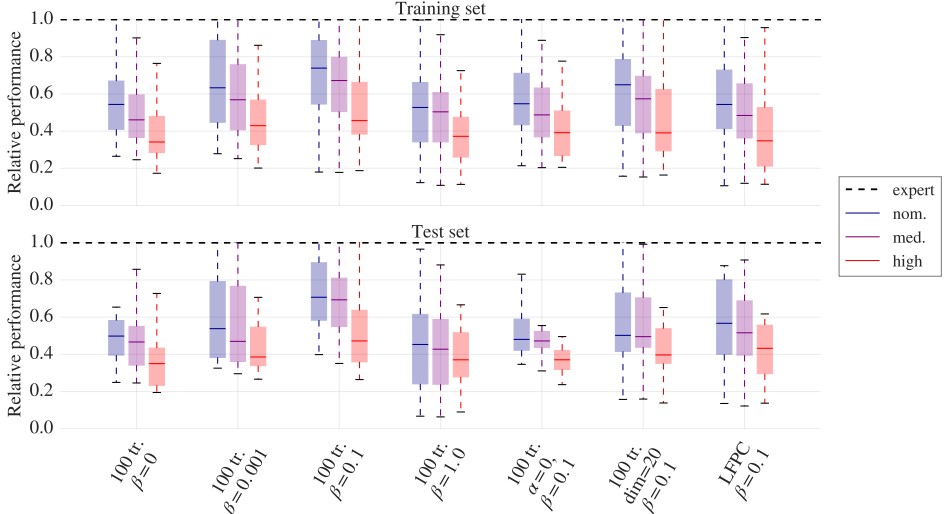

Figure 4: Performance relative to expert policies for trained neural probabilistic motor primitive models. Performance of model variations are compared on training and testing data. We compare models trained using cloning with 100 trajectories per expert for different levels of regularization, using a smaller latent space of dimension 20 rather than 60 in all other experiments, as well as LFPC.

## 3.2 CORE RESULTS: COMPRESSING THOUSANDS OF EXPERTS

Having validated that single skills can be transferred, we next consider how well we can compress behaviors of the 2707 experts in our training set into the neural probabilistic motor primitive architecture. Assessing the models using the action-reconstruction loss is not very intuitive since it does not capture model behavior in the environment. Instead we report a more relevant measure based on expert imitation. Here we encode an expert trajectory into a sequence of latent variables and then *execute the policy in the environment* conditioned on this sequence. Note that this approach is open-loop with respect to the latents while being closed-loop with respect to state. We can then compare the performance of the trained system against experts on training and held-out clips according to the tracking reward used to train the experts originally. To account for different expert reward scales we report performance relative to the expert policy. Importantly, that this approach works is itself a partial validation of the premise of this work, insofar as open-loop execution of action sequences usually trivially fails with minor perturbations. The trained neural probabilistic motor primitive system can execute behaviors conditioned on an open-loop noisy latent variable trajectory, implying that the decoder has learned to stabilize the body during latent-conditioned behavior.

There are a few key takeaways from the comparisons we have run (see Fig. 4). Most saliently cloning based on 100 trajectories from each expert with a medium regularization value ($\beta = 0.1$) works best. LFPC with comparable parameters works less well here, but has qualitatively fairly similar performance. Our ablations show that regularization and a large latent space are important for good results. We also set the autoregressive parameter $\alpha = 0$ (.95 in other runs), making the latent variables i.i.d.. This hurts performance, validating our choice of prior.[4]

## 3.3 ANALYSIS OF THE TRAINED MODEL

We have no expectation that trajectories well outside the training distribution are likely to be either representable by the encoder or executable by the decoder. Nevertheless, when one-shot imitation of a trajectory fails, a natural question is whether the decoder is incapable of expressing the desired actions, or the encoder fails to encode the trajectory in such a way that the decoder will produce it.

---

[4]One other feature of the training pipeline we experimented with is mirroring of policies according to bilateral symmetry (thereby approximately doubling the expert data) – this improves results slightly and all models compared here use mirroring.

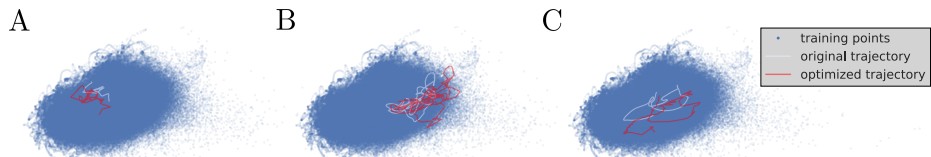

Figure 5: These panels consist of visualizations of the PCA latent space with comparisons in this space between one-shot latent-variable sequences and optimized latent variable sequences for various behaviors: A. Run B. Backwards walking C. Jumping. Running executes well based on the one-shot trajectory so serves as a reference for which optimization is not noticeably different. Walking backwards and jumping one-shot imitations fail, but are noticeably improved by optimization.

We propose an analysis to distinguish this for borderline cases. For held out trajectories that yield unsatisfying performance on one-shot imitation, we can simply optimize directly:

$$\min_{z_1...z_T} \sum_{t=1}^{T} ||\mu_\theta(s_t, z_t) - a_t^\star||_2^2, \tag{10}$$

where $\mu_\theta$ is the decoder mean. Empirically we see that this optimization meaningfully improves the executed behavior, and we visualize the shift in a three-dimensional space given by the first three principal components in Fig. 5.

We exhibit three examples where we visualize the original latent trajectory as well as the optimized latent trajectory. Performance is significantly improved (see supplementary video), showing the latent space can represent behaviors for which one-shot imitation fails. However execution remains imperfect suggesting that while much of the fault may lie with the encoder, the decoder still may be slightly undertrained on these relatively rare behavior categories. Quantitatively, among a larger set of clips with less than 50% relative expert performance for one-shot imitation we found that optimization as described above improved median relative expert performance from 43% to 78%.

Other exploratory probes of the module suggest that it is possible in certain cases to obtain seamless transitioning between behaviors by concatenating latent-variable trajectories and running the policy conditioned on this sequence (e.g. in order to perform a sequence of turns). See additional supplementary video.

**Reuse of motor primitive module** Finally, we experimented with reuse of the decoder as a motor primitive module. We treat the latent space as a new custom action space and train a new high-level (HL) policy to operate in this space. At each time-step the high-level policy outputs a latent-variable

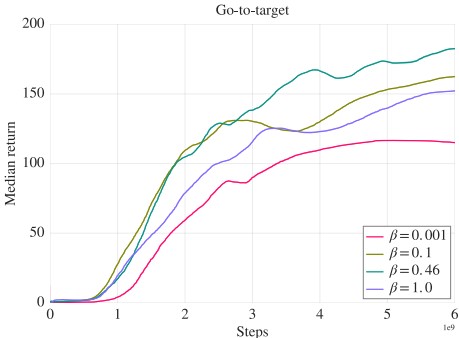
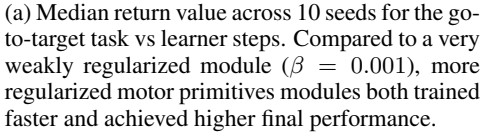

(a) Median return value across 10 seeds for the go-to-target task vs learner steps. Compared to a very weakly regularized module ($\beta = 0.001$), more regularized motor primitives modules both trained faster and achieved higher final performance.

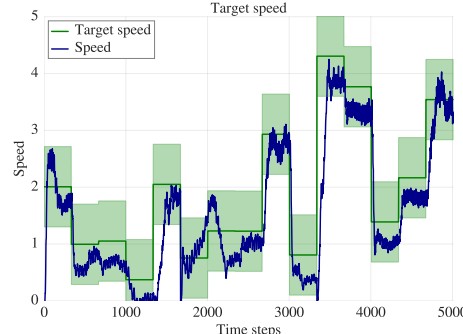

(b) Our model is able to track the target speed accurately. Shown here are target speed and actual speed in the egocentric forward direction for three episodes. The reward function is a Gaussian centered at the target speed. The shaded region corresponds to $\pm$ one standard deviation.

Figure 6: Reuse of neural probabilistic motor primitive modules.

$z_t$. The actual action is then given by the motor primitive module $p(a_t|s_t, z_t)$. For training we used SVG(0) (Heess et al., 2015) with the Retrace off-policy correction for learning the Q-function (Munos et al., 2016).

A natural locomotion task that can challenge the motor module is a task which requires abrupt, frequently redirected movement with sharp turns and changes of speed. To implement this we provide the higher-level controller with a target that is constant until the humanoid is near it for a few timesteps at which point it randomly moves to another nearby location. While no single task will comprehensively probe the module, performing well in this task demands a wide range of quick locomotion behavior. With only a sparse task reward, the HL-controller can learn to control the body through the learned primitive space, and it produces rather humanlike task-directed movement. We observed that more regularized motor primitive modules had more stable initial behavior when connected to the untrained high-level controller (i.e. were less likely to fall at the beginning of training). Compared to a very weakly regularized module ($\beta = 0.001$), more regularized motor primitives modules both trained faster and achieved higher final performance (see Fig. 6a). We also investigated a go-to-target task with bumpy terrain that is unobserved by the agent. The fact that our model can learn to solve this task demonstrates its robustness to unseen perturbations for which the motor primitive module was not explicitly trained. In another experiment we investigated a task in which the agent has to move at a random, changing target speed. This requires transitions between qualitatively different locomotion behavior such as walking, jogging, and running (see Fig. 6b). See an extended video of these experiments. In a final reuse experiment, we consider an obstacle course requiring the agent to jump across gaps (as in Merel et al. (2018)). We were able to solve this challenging task with a high-level controller that operated using egocentric visual inputs (see the main supplementary video).

We emphasize a few points about these results to impact their importance: (1) Using a pretrained neural probabilistic motor primitives module, new controllers can be trained effectively from scratch on sparse reward tasks, (2) the resulting movements are visually rather humanlike without additional constraints implying that the learned embedding space is well structured, and (3) the module enables fairly comprehensive and smooth coverage for the purposes of physics-based control.

## 4    DISCUSSION

In this paper we have described approaches for transfer and compression of control policies. We have exhibited a motor primitive module that learns to represent and execute motor behaviors for control of a simulated humanoid body. Using either a variant of behavioral cloning or linear feedback policy cloning we can train the neural probabilistic motor primitive sytem to perform robust one-shot-imitation, and with the latter we can use relatively restricted data consisting of only single rollouts from each expert. While LFPC did not work quite as well in the full-scale model as cloning from noisy rollouts, we consider it remarkable that it is possible in our setting to transfer expert behavior using a single rollout. We believe LFPC holds promise insofar as it may be useful in settings where rollouts are costly to obtain (e.g. adapted to real-world robotic applications), and there is room for further improvement as we did not carefully tune certain parameters, most saliently the marginal noise distribution $\Delta$.

The resulting neural probabilistic motor primitive module is interpretable and reusable. We are optimistic that this kind of architecture could serve as a basis for further continual learning of motor skills. This work has been restricted to motor behaviors which do not involve interactions with objects and where a full set a of behaviors are available in advance. Meaningful extensions of this work may attempt to greatly enrich the space of behaviors or demonstrate how to perform continual learning and reuse of new skills.

ACKNOWLEDGMENTS

The data used in this project was obtained from mocap.cs.cmu.edu. The database was created with funding from NSF EIA-0196217.

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

## APPENDICES

## A MOTION CAPTURE EXPERTS

The approach we use for producing experts is detailed more fully in Merel et al. (2018). In short, this approach for producing experts largely follows Peng et al. (2018). We took the energy function proposed in SAMCON (Liu et al., 2010), and use it as a per timestep reward to train a time-indexed policy that tracks/imitates a motion capture reference clip (Peng et al., 2018). As proposed in Merel et al. (2017); Peng et al. (2018), episodes are initialized to poses throughout the motion capture reference and episodes are early-terminated when the character falls. Here we use an off-policy RL algorithm, SVG(0) (Heess et al., 2015) with Retrace (Munos et al., 2016). As done in Merel et al. (2017); Peng et al. (2018) and elsewhere, we train stochastic policies and use the mean (i.e. noiseless) action as the expert policy.

## B ARCHITECTURE AND TRAINING DETAILS

The decoder $p(a_t|s_t, z_t)$ in our experiments was a MLP with three layers with 1024 hidden units taking as input the concatenation of state $s_t$ and latent variable $z_t$. The decoder output distribution is a multivariate Gaussian with fixed standard deviation of 0.1 (action values are normalized to $[-1, 1]$). We found that fixing the standard deviation made it significantly easier to prevent overfitting. Note that in this setting varying the $\beta$ parameter is equivalent to varying the fixed output variance (up to a constant). The encoder $q(z_t|z_{t-1}, x_t)$ in our experiments was also an MLP with two layers of 1024 hidden units each. The inputs were simply concatenated at the input. The encoder output distribution was a multivariate Gaussian with learnt variance. In most of our experiments, we used a 60-dimensional latent space.

We used the reparametrization trick (Kingma & Welling, 2013; Rezende et al., 2014) to train the model and used stochastic gradient descent with ADAM (Kingma & Ba, 2015) with a learning rate of 0.0001. In the case of models trained on 100 trajectories per expert we used minibatches of 512 subsequences of length 30.

For LFPC we sampled 32 subsequences of length 30 and produced 5 perturbed state sequences per subsequence. In preliminary experiments the length of the subsequences did not have a major impact on model performance.

## C    RELATIONSHIP TO OTHER KNOWLEDGE TRANSFER IDEAS

Firstly, we note that the emphasis of the proposal in this work is to match the responsivity of the expert policy in a neighborhood around each state. This is distinct from activation matching or KL matching where the emphasis is on matching the action/activation distribution for a particular state (Rusu et al., 2015; Teh et al., 2017). Secondly, we emphasize that the kind of robust knowledge transfer we discuss here is distinct from that which is seen to be important in other settings. For example Srinivas & Fleuret (2018) provide a line of reasoning that involves training a student system to match the exact activations of a teacher in the presence of perturbations on the student inputs. This logic is sound in the setting of large-scale vision systems. However in the context of control policies, this would look like:

$$\min_{\theta} \sum_{s \in \mathcal{S}^{\star}} \mathbb{E}_{\delta s \sim \Delta(s)}[(\mu_E(s) - \mu_\theta(s + \delta s))^2] \tag{11}$$

This essentially means that the student policy is learning to "blindly" reproduce the action of the expert exactly, despite input perturbations. While this is well motivated if the noise is thought to be orthogonal to the proper functioning of the system, this is a very bad idea for control, where you need to pay close attention to small input perturbations. Technically, this amounts to setting the local feedback to zero, and behaving in a sort of *open-loop*-like fashion.

## D    VISUALIZATION OF STATIONARY POLICY BEHAVIOR

Locomotion behavior is, at least in the simplest case roughly a limit cycle. In an additional experiment to test LFPC we gathered three gait cycles of running behavior and performed LFPC. Note that here the student policy need not be time-indexed even when the demonstrations were time-indexed. This restricted case shows striking generalization in the presence of noise (see Fig. A.1 and also see main supplementary video).

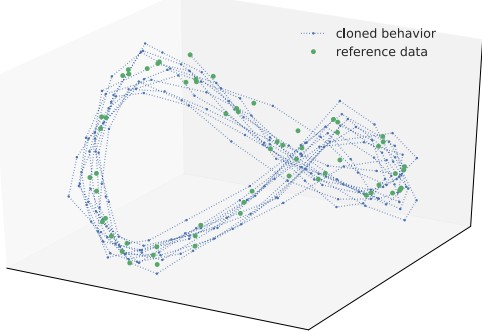

Figure A.1: Dimensionality reduction (PCA) performed on set of poses obtained from noisy rollouts of the stationary cloned policy (blue). The limited reference data originating from a time-indexed policy has been projected into the same space (green). Observe that the rollouts are considerably noisier and consistently deviate from the reference trajectory, nevertheless the cloned-policy trajectories return to the limit cycle.

