# OpenReview forum: "Neural Probabilistic Motor Primitives for Humanoid Control"
_ICLR.cc/2019/Conference_

### Official Review · AnonReviewer1 · 2018-10-16
**Concerns with proposed approach and results**

**Rating:** 4
**Confidence:** 4

**Review:**

This paper considers the problem of transferring motor skills from multiple experts to a student policy. To this end, the paper proposes two approaches: (1) an approach for policy cloning that learns to mimic the (local) linear feedback behavior of an expert (where the expert takes the form of a neural network), and (2) an approach that learns to compress a large number of experts via a latent space model. The approaches are applied to the problem of one-shot imitation from motion capture data (using the CMU motion capture database). The paper also considers an extension of the proposed approach to the problem of high-level planning; this is done by treating the learned latent space as a new action space and training a high-level policy that operates in this space.

Strengths:
S1. The supplementary video was clear and helpful in understanding the setup.
S2. The paper is written in a generally readable fashion.
S3. The related work section does a thorough job of describing the context of the work.

However, I have some significant concerns with the paper. These are described below.

Significant concerns:
C1. My biggest concern is that the paper does not make a strong case for the benefits of LPFC over simpler strategies. The results in Figure 3 demonstrate that a linear feedback policy computed along the expert's nominal trajectory performs as well as (and occasionally even better than) LPFC. This is quite concerning.
C2. Moreover, as the authors themselves admit, "while LPFC did not work quite as well in the full-scale model as cloning from noisy rollouts, we believe it holds promise insofar as it may be useful in rollout-limited settings...". However, the paper does not present any theoretical/experimental evidence that would suggest this.
C3. Another concern has to do with the two-step procedure for LPFC (Section 2.2), where the first step is to learn an expert policy (in the form of a neural network) and the second step is to perform behavior cloning by finding a policy that tries to match the local behavior of the expert (i.e., finding a policy that attempts to produce similar actions as the expert policy linearized about the nominal trajectory). This two-step procedure seems unnecessary; the paper does not make a case for why the expert policies are not chosen as linear feedback controllers (along nominal trajectories) in the first place.
C4. The linearization of the expert policy produced in (1) may not lead to a stabilizing feedback controller and could easily destabilize the system. It is easy to imagine cases where the expert neural network policy maintains trajectories of the system in a tube around the nominal trajectory, but whose linearization does not lead to a stabilizing feedback controller. Do you see this in practice? If not, is there any intuition for why this doesn't occur? If this doesn't occur in practice, this would suggest that the expert policies are not highly nonlinear in the neighborhood of states under consideration (in which case, why learn neural network experts in the first place instead of directly learning a linear feedback controller as the expert policy as suggested in C3?)
C5. I would have liked to have seen more implementation details in Section 3. In particular, how exactly was the linear feedback policy along the expert's nominal trajectory computed? Is this the same as (2)? Or did you estimate a linear dynamical model (along the expert's nominal trajectory) and then compute an LQR controller? More details on the architecture used for the behavioral cloning baseline would also have been helpful (was this a MLP? How many layers?)

Minor comments:
- There are some periods missing at the end of equations (eqs. (1), (2), (6), (8), (9)).

---

> ### Author Response · Authors · 2018-11-14
> **Response to reviewer**
>
> We thank the reviewer for their detailed discussion of the LFPC method and address concerns below. However, as pointed out in the introductory remarks this is only one aspect of the paper and we would also like to encourage the reviewer to include our main result, the neural probabilistic motor primitive module in their assessment.
>
> Addressing the concerns about LFPC in turn:
> C1: For single-behavior experts, indeed we intended Fig 3 to indicate (perhaps surprisingly) that linear-feedback policies perform well, and that LFPC can transfer that level of performance into a new neural network (from a single rollout of behavior).  For a single behavior, this is merely a validation that the new neural network can be as robust as even the linear feedback policy. Our real aim is to be able to distill many experts into a single network as we demonstrate subsequently.
>
> C2: Both LFPC and the behavioral cloning baseline were able to train the NPMP and permit skill reuse, but in our specific one-shot imitation comparisons the behavior-cloning approach performed better.  Behavioral cloning from arbitrary amounts of data is an arbitrarily strong baseline.  The two considerations that motivate LFPC are that we can store fewer data from experts and that we can query fewer trajectories from the expert system (in settings where rollouts are costly, such as real platforms).
>
> C3, C4:  The general setting for our approach is that we assume the existence of experts that perform single behaviors -- as of late, this is a reasonable assumption, enabled by previous research (e.g. Liu et al. 2010, 2015, 2018, Merel et al. 2017, Peng et al. 2018).  What has not been done prior to this work is to exhibit single policies capable of flexibly generating a wide range of skills, and this is the problem we are focusing on.  For our purposes, it is not critical how experts are obtained, and this paper does not advocate any particular way of generating expert policies.  That being said, neural network experts have been successfully trained in some recent work, so we expected it would work, and a priori, it was not obvious that directly training a linear feedback policy might suffice.  Moreover, in preliminary experiments done when beginning this work (not reported here), we found that it can be quite data inefficient to directly train a time-indexed linear feedback policy for tracking motion capture using RL, we believe due to lack of parameter sharing across timesteps, so we did not pursue this further.
>
> Nevertheless our single-behavior expert transfer experiments demonstrated empirically that linear feedback policies extracted from the expert neural networks were essentially as performant as RL-trained neural network experts in terms of robust tracking of single behaviors (Fig. 3).  That linear feedback policies work as well here is a statement about the dynamics of the environment and the complexity of the behaviors (i.e. that the behaviors here are sufficiently unimodal).  It seems, for a wide range of stereotyped behaviors, the policies required to execute the behaviors might be “surprisingly simple”, depending on your initial preconceptions.
>
> C5: In contemporary neural network languages, it is straightforward to compute the Jacobian of the actions w/ respect to observation inputs.  As described in eqn 2, this directly provides the linearization of the policy when evaluated at the nominal trajectory.  In section 3.1, we use the same network architecture for cloning from noisy rollouts and from the linear feedback policy (MLP with two hidden layers: 1024, 512; we can add this detail to the text).
>
>
> References:
> Liu, L., Yin, K., van de Panne, M., Shao, T. and Xu, W., 2010. Sampling-based contact-rich motion control. ACM Transactions on Graphics (TOG), 29(4), p.128.
>
> Liu, L., Yin, K. and Guo, B., 2015, May. Improving Sampling‐based Motion Control. In Computer Graphics Forum (Vol. 34, No. 2, pp. 415-423).
>
> Libin Liu and Jessica Hodgins. Learning basketball dribbling skills using trajectory optimization and deep reinforcement learning. ACM Transactions on Graphics (TOG), 37(4):142, 2018.
>
> Merel, Josh, Yuval Tassa, Sriram Srinivasan, Jay Lemmon, Ziyu Wang, Greg Wayne, and Nicolas Heess. "Learning human behaviors from motion capture by adversarial imitation." arXiv preprint arXiv:1707.02201 (2017).
>
> Xue Bin Peng, Pieter Abbeel, Sergey Levine, and Michiel van de Panne. Deepmimic: Example-guided deep reinforcement learning of physics-based character skills. arXiv preprint arXiv:1804.02717, 2018.

---

### Official Review · AnonReviewer2 · 2018-11-04
**Sound approach, but very similar to prior work**

**Rating:** 6
**Confidence:** 4

**Review:**

The paper tackles the problem of distilling large numbers of expert demonstrations into a single policy that can both recreate original demonstrations in a physically-simulated environment and humanoid platform, and to generalize to novel motions. Towards this, the paper presents two approaches learn policies from expert demonstrations without involving costly closed loop RL training, and distilling these individual experts into a shared policy by learning latent time-varying codes.

The paper is well-written and the method is well-evaluated in the scope that it is proposed. Both components of the proposed approach have previously been explored in the literature - there is extensive work on learning local controllers for physics based evironments from demonstrations in both open loop and closed loop settings as well as work on mixtures of these controllers in machine learning, robotics and computer graphics communities. While the paper proposes these two components as a contribution, I would like to see a more detailed argument of what this work contributes over previous such approaches.

Another part  where I wish the paper could make a more compelling argument is that distilled policy can perform non-trivial generalization. Target following is a good illustrative example, but has been showcased by multitude of prior work. The paper talks about compositionality, and it would have been compelling to see examples of that if the method can achieve it. For example, simultaneously performing locomotion skills with upper body manipulation skills is something mixture of expert demonstrations approaches still struggle with and it would have been great to see this paper investigate the approach on this problem.

Overall, this is a sound and well-written submission, but the existence of very related prior work with similar capabilities makes me reluctant to recommend this paper.

---

> ### Author Response · Authors · 2018-11-14
> **Response to reviewer**
>
> We thank the reviewer for appreciating the difficult problem we’re tackling.  However, we disagree with the reviewer about the level of similarity between this work and previous work.  We have discussed a number of relationships between this work and existing approaches in the robotics, ML, and graphics communities. As far as we are aware, no existing work learns a rich embedding space for physics-based control.  For kinematic sequence modeling, there is abundant work in computer graphics that learns to blend/reuse/compose movement trajectories (e.g. Holden et al. 2017).  To our knowledge, for the much more challenging problem of flexible physics-based control, there is no prior work which results in a robustly reusable skill space that is as comprehensive in scope as what was demonstrated here.  We would sincerely appreciate references of any previous papers that the reviewer thinks overlap in terms of successfully demonstrating the learning of a skill space which is reusable for physics-based control, especially for humanoids.
>
> One-shot imitation has been demonstrated by a few groups in the past couple years for mounted robotic arms. But we are aware of considerably less work (primarily Wang et al. 2017; discussed in the paper) in which humanoids perform one-shot behaviors.  The reason this is difficult in the physics-based case is that the humanoid must balance and remain upright in addition to imitating the demonstration.  Moreover, while one-shot imitation is the core systematic test of the model, since the architecture was trained for this setting, we emphasize that the demonstration of reuse is considerably more interesting to us.  After producing this module, a fresh HL policy can learn to set the “intention” of the LL controller and produces fairly human-like behaviors by reusing the learned skill space.  We selected the go-to-target task because we wanted to heavily tax the LL movement space by demanding sudden, jerky changes of movement and what resulted were strikingly human-like movement changes, with only a very simple reward (reward = 0 everywhere except when target is reached) and no additional constraints on the human-likeness of the behavior.  While simpler bodies can solve this problem from scratch, for a complex humanoid, the movements produced by learning from scratch are most definitely very non-human-like in general.
>
> Simultaneously reusing the upper body for manipulation while having lower body locomote is indeed a great challenge problem for future work. We have already included imitation of arm movements in our evaluation but our training distribution does not contain any manipulation demonstrations. We are optimistic that this approach can scale to this setting, but it is beyond the scope of the present paper.  We do believe, that what we have demonstrated here advances the state of the art for reusable physics-based locomotion behaviors.
>
> References:
> Holden, Daniel, Taku Komura, and Jun Saito. "Phase-functioned neural networks for character control." ACM Transactions on Graphics (TOG) 36, no. 4 (2017): 42.
>
> Ziyu Wang, Josh S Merel, Scott E Reed, Nando de Freitas, Gregory Wayne, and Nicolas Heess. Robust imitation of diverse behaviors. In Advances in Neural Information Processing Systems, pp. 5320–5329, 2017.

---

### Official Review · AnonReviewer3 · 2018-11-05
**The idea is oversimplified, which may limit its applications.**

**Rating:** 3
**Confidence:** 4

**Review:**

This paper mainly focuses the imitation of expert policy as well as compression of expert skills via a latent variable model. Overall, I feel this paper is not quite readable, albeit that the prosed methods are simple and straightforward.

As one major contribution of this paper, the authors introduce a first-order approximation to estimate the action of an expert, where perturbations are considered. However, this linear treatment could yield large errors when the residuals in (1) are still large, which is very common in high-dimensional and highly-nonlinear cases. Specifically, the estimation of “J” could be hard. In addition, just below (1), the authors mention (1) yields a “stabilized policy”, so what do you mean “stabilized”?

Another crucial issue lies on the treatment of “\Delta(s)”, which is often unknown and hard to modeled, Thus, various optimal controllers are introduced so as to obtain robust controllers. Similarly, in (9) it is also difficult to decide what is “suitable perturbation distribution”.

Overall, the linear treatment in (2) and assumption on “\Delta(s)” in (5) actually oversimplify the imitation learning problem, which may not be applicable in real robot applications.

Others small comments:
-Section 2.1 could be moved to supplementary material or appendix, as this part is indeed not a contribution.

- in (5), it should be “-J_{i}^{*}”

---

> ### Author Response · Authors · 2018-11-14
> **Response to reviewer**
>
> Concerning LFPC, we note that bipedal locomotion is highly nonlinear and despite this, the linear feedback policy empirically works rather robustly (despite the high-D observation space) as shown in section 3.1.  The term linear-feedback-stabilized policy, refers to the linear feedback policy in equation 2, which is stabilized with linear feedback (relative to the naive open-loop policy that simply executes a fixed sequence of actions).
>
> We consider it clear from our results that time-indexed linear feedback policies suffice to capture the behavior of experts around nominal trajectories in our setting. Correspondingly, LFPC is capable of transferring expert functionality. We would like to point out that in our scenario there is no need to estimate J -- it is simply the Jacobian of a neural network with respect to the inputs which is readily available in standard neural network languages (see eqn 2).
>
> There seems to be some confusion about delta s -- it has very little to do with the “various optimal controllers” and indeed we state in the paper (page 4) that the approach is fairly insensitive to precise selection of this distribution. One possible reason for this is that the distribution does not matter much as long as it covers the states visited by the linear feedback policy which appears to stay pretty close to the nominal trajectory.
>
> Finally, the reviewer expresses concerns with respect to the applicability of our approach to the real robot setting. Our paper primarily targets the control of simulated physical humanoids and we do not make any further claims.  However recent approaches in a similar imitation learning setting have been shown to be effective for real robots (e.g. Laskey et al. 2017), so we do believe, as we speculate in the discussion that this is a plausible direction for future work.
>
> We thank the reviewer for spotting a typo in equation 5 which we will correct.
>
> References:
> Laskey, M., Lee, J., Fox, R., Dragan, A. and Goldberg, K., 2017. Dart: Noise injection for robust imitation learning. arXiv preprint arXiv:1703.09327.

---

### Author Response · Authors · 2018-11-14
**Overall response to reviewers**

We thank all reviewers for their time and comments.

We would like to emphasize that there are two contributions in the work.  The focal motivation is the production of a single trained motor architecture which can execute and reuse motor skills of a large, diverse set of experts with minimal manual segmentation or curation.  The architecture that we develop permits one-shot imitation as well as reuse of low-level motor behaviors in the context of new tasks.

Our main results involve one-shot imitation and motor reuse, using our trained module for a humanoid body with relatively high action DoF.  We believe this novel architecture enables more generic behavior and motor flexibility than other work involving learning to control physically simulated humanoids.

AnonReviewer3 and AnonReviewer1 essentially restrict criticism of the work to the LFPC approach, which is only one aspect of our research contribution. We address these concerns in detail below. But we would also encourage the reviewers to assess the quality and novelty of the core architectural contributions as well as the quality of the experimental results.  We are not aware of previous work for control of a physically simulated humanoid that demonstrates a learned module that can execute many behavioral skills and permits reuse.

---

### Author Response · Authors · 2018-11-16
**Revised draft posted.**

In response to reviewer feedback, we have revised our abstract and contributions portion of the introduction to better communicate the focus of the paper.  We consider the neural probabilistic motor primitive module to be the primary contribution and LFPC as an auxiliary contribution.  As judged by reviewer reception, this did not come across as intended.  We hope the revision better reflects this.

---

### Comment · Area_Chair1 · 2018-11-19
**replies from reviewers to author responses?**

This paper has seen detailed reviews and detailed responses by the authors. Thank you to all.

Reviewers:  please do provide further feedback based on the authors replies,
and note whether it changes your evaluation and your score for the paper.
Also note that a revised draft has been submitted.
Your input is greatly appreciated, as the opinions are mixed and they focus on different aspects of the work.

For revision differences of the revised draft:
select "Show Revisions" on the review page, and then select the check-boxes for the versions you wish to compare.

-- area chair

---

### Comment · AnonReviewer1 · 2018-11-20
**Response to revised version of paper**

I have read all of the comments (from the reviewers and the authors) and have also read the revised version of the paper. I am still not convinced that the paper makes a strong contribution. Here are my comments:

- The revised version of the paper still has LPFC as a major portion of the paper. In particular, the real estate in terms of pages devoted to explaining LPFC is more than that devoted to neural probabilistic motor primitives (which the authors claim is the main contribution of the paper). The conclusion of the paper also highlights LPFC (including its limitations). I do not think that the revised version of the paper adequately de-emphasizes LPFC.

- The results from the simulation experiments showcasing neural probabilistic motor primitives (NPMP) presented in the paper are not particularly compelling. In particular, Figure 4 (which presents the relative performance of NPMP as compared to the expert) suggests that NPMP is not really doing a good job at capturing the expert's behavior. In particular, for both training and test data, the relative performance is around 0.5, which doesn't seem particularly good. Moreover, as noted by AnonReviewer2, the target following example is not particularly compelling, since it has previously been demonstrated by many other papers. I would thus have liked to have seen a thorough comparison of NPMP with other methods on this example. Moreover, as noted in my previous review, the results for LPFC are also quite weak.

Based on this, I retain my original rating for the paper.

Small comments:
- For clarity, I would recommend using \eqref{} when referencing equations. For example, on pg. 6, "Objective 5" should be "Objective (5)".

---

### Comment · Area_Chair1 · 2018-12-08
**final remarks?  R3?**

We are reaching the end of the discussion period.
There remain mixed opinions on the paper.
Any further thoughts from R2 and R3? Stating pros + cons and summarizing any change in opinion would be very useful.
The main contribution is centred around one-shot imitation as well as reuse of low-level motor behaviors in the context of new tasks. Issues being discussed include related prior art,  demonstrated benefit of method in results, importance of LFPC.
Of course we recognize that reviewer & author time is limited.
-- area chair

---

### Meta-Review · Area_Chair1 · 2018-12-14
**reviews on balance lean negative, but recommend accept  (is this excessive influence of the AC opinion?)**

**Confidence:** 2
**Recommendation:** Accept (Poster)

**Metareview:**

Strengths:  One-shot physics-based imitation at a scale and with efficiency not seen before.
Clear video, paper, and related work.

Weaknesses described include:  the description of a secondary contribution (LFPC)
takes up too much space (R1,4); results are not compelling (R1,4); prior art in graphics and robotics (R2,6);
concerns about the potential limitations of the linearization used by LFPC.

The original reviews are negative overall (6,3,4). The authors have posted detailed replies.
R1 has posted a followup, standing by their score. We have not heard more from R2 and R3.

The AC has read the paper, watched the video, and read all the reviews.
Based on expertise in this area, the AC endorses the author's responses to R1 and R2.
Being able to compare LFPC to more standard behavior cloning is a valuable data point for the community;
there is value in testing simple and efficient models first.
The AC identifies the following recent (Nov 2018) paper as being the closest work, which is not identified by the authors or the reviewers. The approach being proposed in the submitted paper demonstrates equal-or-better scalability,
learning efficiency, and motion quality, and includes examples of learned high-level behaviors.
An elaboration on HL/LL control: the DeepLoco work also learns mocap-based LL-control with learned HL behaviors.
       although with a more dedicated structure.
       Physics-based motion capture imitation with deep reinforcement learning
       https://dl.acm.org/citation.cfm?id=3274506

Overall, the AC recommends this paper to be accepted as a paper of interest to ICLR.
This does partially discount R3 and R1, who may not have worked as directly on these specific problems before.

The AC requests is rating the confidence as "not sure" to flag this for the program committee chairs, in light of the fact that this discounts the R1 and R3 reviews.
The AC is quite certain in terms of the technical contributions of the paper.